# *GREB1* amplifies androgen receptor output in human prostate cancer and contributes to antiandrogen resistance

Eugine Lee[1], John Wongvipat[1], Danielle Choi[1], Ping Wang[2], Young Sun Lee[1], Deyou Zheng[2,3,6], Philip A Watson[1], Anuradha Gopalan[4], Charles L Sawyers[1,5]*

[1]Human Oncology and Pathogenesis Program, Memorial Sloan Kettering Cancer Center, New York, United States; [2]Department of Genetics, Albert Einstein College of Medicine, New York, United States; [3]Department of Neurology, Albert Einstein College of Medicine, New York, United States; [4]Department of Pathology, Memorial Sloan Kettering Cancer Center, New York, United States; [5]Howard Hughes Medical Institute, Chevy Chase, United States; [6]Department of Neuroscience, Albert Einstein College of Medicine, New York, United States

**Abstract** Genomic amplification of the androgen receptor (*AR*) is an established mechanism of antiandrogen resistance in prostate cancer. Here, we show that the magnitude of *AR* signaling output, independent of *AR* genomic alteration or expression level, also contributes to antiandrogen resistance, through upregulation of the coactivator *GREB1*. We demonstrate 100-fold heterogeneity in *AR* output within human prostate cancer cell lines and show that cells with high *AR* output have reduced sensitivity to enzalutamide. Through transcriptomic and shRNA knockdown studies, together with analysis of clinical datasets, we identify *GREB1* as a gene responsible for high *AR* output. We show that *GREB1* is an *AR* target gene that amplifies *AR* output by enhancing *AR* DNA binding and promoting *EP300* recruitment. *GREB1* knockdown in high *AR* output cells restores enzalutamide sensitivity *in vivo*. Thus, *GREB1* is a candidate driver of enzalutamide resistance through a novel feed forward mechanism.
DOI: https://doi.org/10.7554/eLife.41913.001

*For correspondence:
sawyersc@mskcc.org

## Introduction

Androgen receptor (*AR*) targeted therapy is highly effective in advanced prostate cancer but is complicated by the emergence of drug resistance, called castration-resistant prostate cancer (CRPC) (*Shen and Abate-Shen, 2010*; *Watson et al., 2015*). The most common mechanism of CRPC is restored *AR* signaling, primarily through amplification of *AR* (*Chen et al., 2004*; *Robinson et al., 2015*). The importance of *AR* amplification as a clinically important drug resistance mechanism is underscored by recent data showing that *AR* amplification, detected in circulating tumor DNA or in circulating tumor cells (CTCs), is correlated with reduced clinical benefit from the next generation *AR* inhibitors abiraterone or enzalutamide (*Annala et al., 2018*; *Podolak et al., 2017*).

Genomic landscape studies of prostate cancer have revealed several molecular subtypes defined by distinct genomic drivers (*Berger et al., 2011*; *Cancer Genome Atlas Research Network, 2015*; *Taylor et al., 2010*). In addition to this genomic heterogeneity, primary prostate cancers also display heterogeneity in *AR* transcriptional output, measured by an *AR* activity score (*Hieronymus et al., 2006*). Notably, these differences in transcriptional output occur in the absence of genomic alterations in *AR*, which are generally found only in CRPC (*Cancer Genome Atlas Research Network, 2015*). One potential explanation for this heterogeneity in *AR* transcriptional output is through coactivators and other *AR* regulatory proteins such as *FOXA1*, *SPOP*, *FOXP1* and *TRIM24*

**eLife digest** The prostate is a roughly walnut-sized gland that makes up part of the reproductive system in men. The normal development of this gland depends on a protein known as the androgen receptor. This protein also plays an important role in driving the growth of prostate cancers, and doctors routinely treat such cancers with drugs that block the androgen receptor. While these treatments often shrink the tumors at first, the prostate cancer cells commonly become resistant to the existing "antiandrogen" drugs, highlighting the need to find new drugs for this cancer.

The main way that prostate cancers become resistant to antiandrogen drugs is by making more of the androgen receptor. As such, a better understanding of this protein's activity may prove vital to discovering new treatments. Together with other proteins called co-factors, the androgen receptor binds to DNA and switches on a set of target genes that drive the growth of prostate cancers. The activity of these genes, referred to as "androgen receptor output", varies between different patients with prostate cancer and even between different cells from a single patient's tumor. This variation may occur even when the level of the androgen receptor is constant, which suggests that other factors affect the output of the androgen receptor.

Lee et al. set out to discover if cells with different androgen receptor outputs, but constant androgen receptor levels, respond differently to antiandrogen drugs. First, human prostate cancer cells were separated according to their androgen receptor output. Lee et al. then treated all the cells with an antiandrogen drug known as enzalutamide: tumors grown from cells with a high output became resistant to the drug faster than cells with low output. Next, a large-scale experiment revealed the differences in gene activity between cells with high and low outputs. On average, the cells with a high androgen receptor output had more of an androgen receptor co-factor called GREB1 than the cells with a low output. Biochemical experiments showed that the GREB1 protein interacts with the androgen receptor and amplifies the expression of the receptor's target genes. When the levels of the GREB1 protein were experimentally decreased in prostate cancer cells with a high androgen receptor output, the cells became less resistant to the antiandrogen drug.

Future work will be needed to know if GREB1 levels are a good proxy for patients with high androgen receptor output. The current work predicts that those patients will respond less well to current antiandrogen drugs. A better understanding of how GREB1 and androgen receptor cooperate may also be useful for developing new drugs to treat prostate cancer.

DOI: https://doi.org/10.7554/eLife.41913.002

(*Cancer Genome Atlas Research Network, 2015*; *Geng et al., 2013*; *Groner et al., 2016*; *Pomerantz et al., 2015*; *Takayama et al., 2014*).

Much of the work to date has focused on inter-tumoral heterogeneity. Here, we address the topic of intra-tumoral heterogeneity in *AR* transcriptional output, for which we find substantial evidence in prostate cancer cell lines and in primary prostate tumors. Using a sensitive reporter of *AR* transcriptional activity to isolate cells with low versus high *AR* output, we show that high *AR* output cells have an enhanced response to low doses of androgen and reduced sensitivity to enzalutamide, in the absence of changes in *AR* mRNA and protein expression. To understand the molecular basis for these differences, we performed transcriptome and shRNA knockdown studies and identified three genes (*GREB1*, *KLF8* and *GHRHR*) upregulated in high *AR* output cells, all of which promote *AR* transcriptional activity through a feed-forward mechanism. Of these, we prioritized *GREB1* for further characterization because *GREB1* mRNA levels are increased in primary prostate tumors that have high *AR* activity. *GREB1* amplifies *AR* transcriptional activity through a two-part mechanism: by promoting *EP300* recruitment and by enhancing *AR* binding to chromatin. Importantly, *GREB1* knockdown converted high *AR* output cells to a low *AR* output state and restored enzalutamide sensitivity *in vivo*. Collectively, these data implicate *GREB1* as an *AR* signal amplifier that contributes to prostate cancer disease progression and antiandrogen resistance.

## Results

### Isolation of cells with low and high *AR* output but comparable *AR* expression

Previous work using a *KLK3* promoter/GFP reporter (PSAP-eGFP) showed that LNCaP prostate cancer cells display varying levels of eGFP expression. Characterization of low GFP cells in this analysis revealed reduced *AR* levels and increased expression of stem cell and developmental gene sets (*Qin et al., 2012*). We explored this question in the context of the contemporary data on heterogeneity in *AR* transcriptional output using a different *AR*-responsive reporter, ARR$_3$tk-eGFP, where eGFP expression is driven by the probasin promoter modified to contain three *AR* responsive elements (*Snoek et al., 1998*). LNCaP (*Figure 1*) and CWR22PC-EP (*Figure 1—figure supplement 1*) prostate cancer cells containing a single copy of the reporter construct were derived by infection with lentivirus containing the reporter at a low multiplicity of infection (MOI) (*Figure 1A*). Remarkably, we observed >100 fold range in eGFP expression, as measured by flow cytometry, despite similar levels of *AR* by immunofluorescence microscopy (*Figure 1B,C*, *Figure 1—figure supplement 1A*).

We then used flow cytometry to isolate eGFP-positive cells from both ends of the spectrum of *AR* transcriptional output, which we refer to as ARsig-hi (high *AR* output) and ARsig-lo (low *AR* output) cells, respectively (*Figure 1C*, *Figure 1—figure supplement 1A*). ARsig-hi cells also express higher levels of endogenous *AR* target genes (*FKBP5*, *KLK3*, *TRPM8*) (*Figure 1D,E*, *Figure 1—figure supplement 1B,C*), and have an overall increase in *AR* transcriptional activity based on RNA-sequencing analysis (*Figure 1F*). In addition, the ARsig-lo and ARsig-hi transcriptional phenotypes remain stable for over 30 days post-sorting (*Figure 1G*, *Figure 1—figure supplement 1D*). Interestingly, ARsig-lo cells showed upregulation of gene sets related to proliferation and cell cycle (*Figure 1—source data 1*). Of note, *Qin et al. (2012)* reported downregulation of these gene sets in their low/absent *KLK3* cells, suggesting that the two reporters read out different transcriptional activities. Importantly, the difference in *AR* output between ARsig-lo and ARsig-hi cells is not explained by different levels of *AR* expression or nuclear translocation, since both were comparable in each subpopulation (*Figure 1D,E*, *Figure 1—figure supplement 1B,C*, *Figure 1—figure supplement 2*).

We next asked if isolated ARsig-lo and ARsig-hi populations have different responses to ligands such as dihydrotestosterone (DHT) or antagonists such as enzalutamide. ARsig-hi cells showed enhanced sensitivity to DHT in a dose-dependent manner (*Figure 1H*; *Figure 1—figure supplement 1E*). This result is similar to the effect of increased *AR* expression in conferring sensitivity to low doses of androgen (*Chen et al., 2004*), but now without a change in *AR* level. To address sensitivity to enzalutamide, we used LNCaP/AR xenografts (derived from LNCaP cells) because this model has a track record of revealing clinically relevant mechanisms of enzalutamide resistance (*Arora et al., 2013*; *Balbas et al., 2013*). As we did with LNCaP and CWR22PC-EP cells, we derived ARsig-lo and ARsig-hi subpopulations by flow cytometry and also observed differential *AR* output despite similar levels of *AR* expression (*Figure 1—figure supplement 3A–C*). Remarkably, ARsig-hi cells developed enzalutamide resistance significantly faster that ARsig-lo or parental cells when injected into castrated mice treated with enzalutamide (*Figure 1I*).

Having demonstrated heterogeneous *AR* output within prostate cancer cell lines, we asked if similar, intra-tumoral heterogeneity is observed clinically by immunohistochemical analysis of *KLK3* and *AR* expression in several primary cancers. Consistent with previous reports (*Qin et al., 2012*; *Ruizeveld de Winter et al., 1994*), we observed heterogeneous *KLK3* staining that is not strictly correlated with *AR* level. For example, we found variable intensity of *KLK3* staining in tumor cells with comparable levels of *AR* staining (lined boxes; *Figure 1—figure supplement 4*) and, conversely, variable intensity of *AR* staining in tumor cells with similar *KLK3* staining (dotted circles; *Figure 1—figure supplement 4*). Although this is a small dataset, the results indicate that the *AR* transcriptional heterogeneity we observe in prostate cancer cell lines is present in patient samples. Emerging technologies for conducting single cell RNA and protein analysis in clinical material will enable deeper investigation of this question.

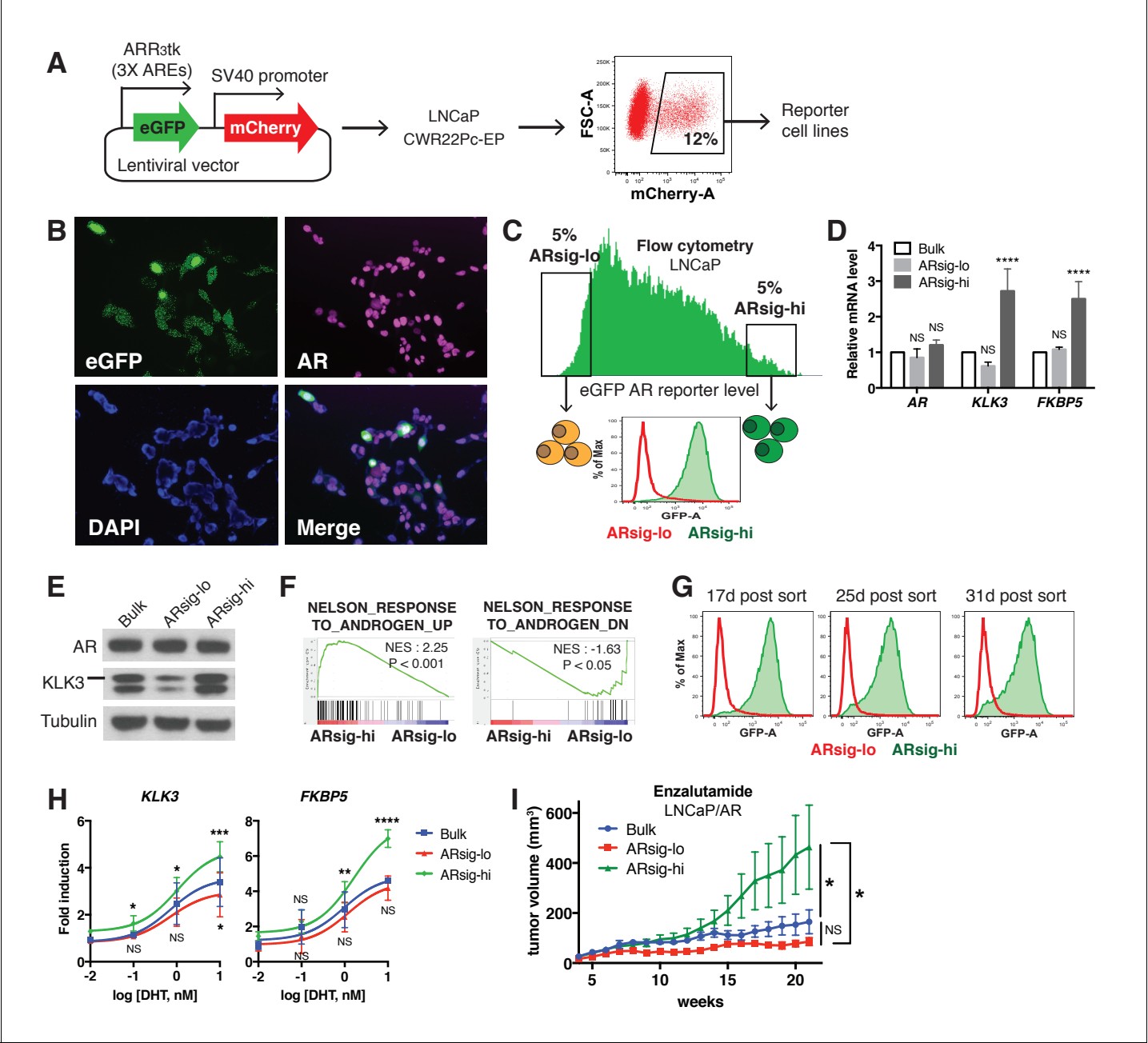

**Figure 1.** Characterization of prostate cancer cells with low vs. high *AR* output. (**A**) The LNCaP and CWR22Pc-EP reporter cell lines were generated by lentiviral infection with the eGFP *AR* reporter construct (details can be found in Materials and methods). Cells with stable integration of the construct were positively sorted by mCherry expression using flow cytometry. (**B**) LNCaP cells infected with the *AR* reporter display variable expression levels of eGFP (green) and AR (magenta). Nuclei were labeled with DAPI (blue). (**C**) LNCaP cells with low (ARsig-lo) or high (ARsig-hi) *AR* activities were sorted using flow cytometry based on eGFP *AR*-reporter expression. (**D–E**) LNCaP ARsig-hi cells have higher *AR* output while having the same level of *AR*. The q-PCR data (**D**) is presented as mean fold change ±SD relative to the bulk population. NS = not significant, ****p<0.0001, one-way ANOVA compared to the bulk population. (**F**) Gene set enrichment analysis (GSEA) shows that the gene sets up- and down-regulated by androgen are enriched in LNCaP ARsig-hi and ARsig-lo cells, respectively. (**G**) LNCaP ARsig-lo and ARsig-hi cells maintain their *AR* activity levels over time. (**H**) LNCaP ARsig-hi cells showed enhanced upregulation of *AR* target genes in response to DHT treatment. The q-PCR data is presented as mean fold change ±SD relative to the DMSO control. NS = not significant, *p<0.05, **p<0.01, ***p<0.001, ****p<0.0001, one-way ANOVA compared to the bulk population. (**I**) LNCaP/AR xenografts derived from ARsig-hi cells become resistant to enzalutamide faster than other populations. The bulk, sorted ARsig-lo and ARsig-hi cells were injected into physically castrated mice and the mice were treated with enzalutamide immediately after injection. Data is presented as mean ±SEM (N = 10). NS = not significant, *p<0.05, one-way ANOVA.

DOI: https://doi.org/10.7554/eLife.41913.003

*Figure 1 continued on next page*

*Figure 1 continued*

The following source data and figure supplements are available for figure 1:

**Source data 1.** GSEA Results (ARsig-lo vs. ARsig-hi).
DOI: https://doi.org/10.7554/eLife.41913.008
**Figure supplement 1.** Characterization of CWR22Pc-EP prostate cancer cells with low vs. high *AR* output.
DOI: https://doi.org/10.7554/eLife.41913.004
**Figure supplement 2.** LNCaP ARsig-lo and ARsig-hi cells have comparable nuclear AR levels.
DOI: https://doi.org/10.7554/eLife.41913.005
**Figure supplement 3.** Characterization of LNCaP/AR prostate cancer cells with low vs. high *AR* output.
DOI: https://doi.org/10.7554/eLife.41913.006
**Figure supplement 4.** AR and KLK3 staining in untreated localized prostate cancer shows heterogeneous KLK3 staining that is not strictly correlated with AR level.
DOI: https://doi.org/10.7554/eLife.41913.007

## GREB1 maintains high AR transcriptional output

To elucidate the molecular basis underlying the differences in ARsig-lo and ARsig-hi cells, we performed RNA-sequencing and found 69 genes upregulated in ARsig-lo cells and 191 genes upregulated in ARsig-hi cells (fold change ≥1.5, p<0.05, *Figure 2—source data 1*). In addition to enrichment of gene sets regulated by androgen (*Figure 1F*), human prostate luminal and basal cell gene sets were enriched in ARsig-hi and ARsig-lo cells, respectively (*Figure 2A*). Based on these results, we postulated that high *AR* output could be a consequence of upregulation of transcriptional co-activators and/or of genes involved in luminal differentiation. We therefore filtered the list of 191 genes upregulated in ARsig-hi cells and identified 33 genes annotated as co-activators or luminal genes (*Figure 2—source data 2*), then measured the consequence of shRNA knockdown of each one on *AR* output in ARsig-hi cells (*Figure 2B*). Three of the 33 candidate genes (*GREB1*, *GHRHR*, *KLF8*) inhibited *AR* activity when knocked down in ARsig-hi cells, with successful knockdown confirmed by qRT-PCR (*Figure 2C,D*). *AR* knockdown served as a positive control, and *ACPP* (one of the 30 genes that did not score) served as a negative control. Interestingly, all three hits are transcriptional upregulated by DHT simulation (*Figure 2E*), which likely explains their increased expression in ARsig-hi cells.

Among the three, *GREB1* emerged as the most compelling candidate for further investigation based on interrogation of clinical datasets. Specifically, we found a statistically significant positive correlation (*r*) between *GREB1* RNA level and *AR* output score (*Cancer Genome Atlas Research Network, 2015*; *Hieronymus et al., 2006*) across the primary prostate tumors from the TCGA dataset, but not *GHRHR* or *KLF8* (*Figure 2F*). Consistent with this, increased expression of *GREB1*, but not *GHRHR* or *KLF8*, was observed in TCGA cases with high *AR* scores (top 5%) versus low *AR* scores (bottom 5%) (*Figure 2F*, *Figure 2—source data 3*). To be sure that *GREB1* is relevant in other model systems, we confirmed *GREB1* upregulation in CWR22PC-EP ARsig-hi cells (*Figure 2—figure supplement 1A*) and reduced *AR* output after *GREB1* knockdown (*Figure 2—figure supplement 1B*). We further validated the knockdown data using CRISPR/Cas9, which also showed inhibition of *AR* output (by flow cytometry) and highly reduced *KLK3* expression in LNCaP ARsig-hi sublines expressing different sgRNAs targeting *GREB1*, without detectable changes in *AR* protein level (*Figure 2G,H*).

## GREB1 amplifies AR transcriptional activity by enhancing AR DNA binding

*GREB1* was first reported as an estrogen-regulated gene in breast cancer (*Rae et al., 2005*) then shown to bind directly to ER, presumably through its LxxLL motif, and function as an ER coactivator by promoting interaction with cofactors (*Mohammed et al., 2013*). To determine if *GREB1* also functions as an *AR* coactivator, we introduced exogenous *GREB1* (HA-GREB1) into ARsig-lo LNCaP and CWR22PC-EP cells and derived stably expressing sublines (*Figure 3A*, *Figure 3—figure supplement 1A*). *GREB1* overexpression enhanced DHT-induced *AR* target gene expression in a dose-dependent manner (*Figure 3B,C*, *Figure 3—figure supplement 1B*), indicating that *GREB1* also promotes *AR* activity.

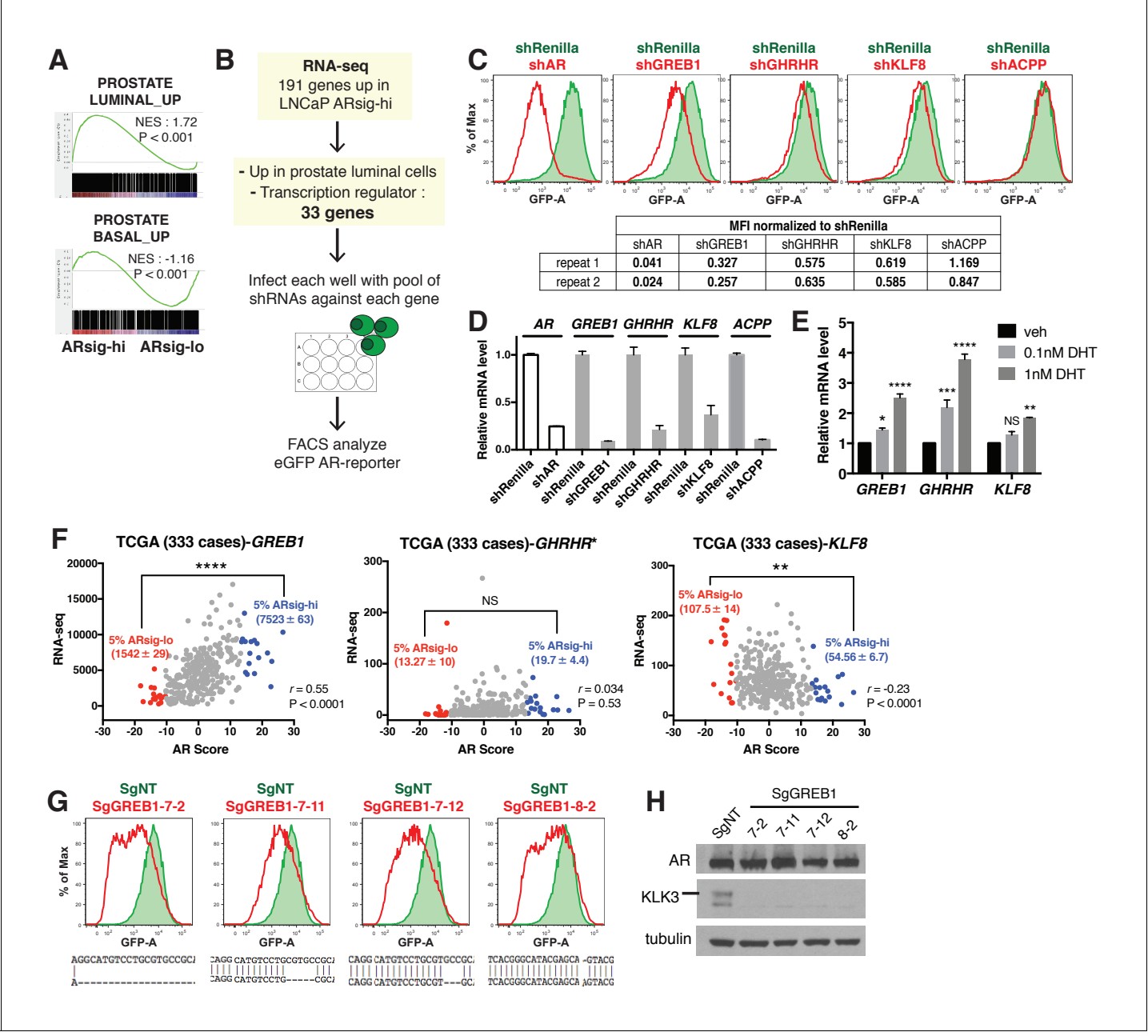

**Figure 2.** Knockdown of the three *AR* regulated genes, *GREB1*, *GHRHR* and *KLF8*, inhibited *AR* activity in cells with high *AR* activity. (**A**) Gene set enrichment analysis (GSEA) shows that genes upregulated in human prostate luminal and basal cells are enriched in LNCaP ARsig-hi and ARsig-lo cells, respectively. (**B**) The schematic of the knockdown study with 33 selected genes upregulated in LNCaP ARsig-hi cells. Details can be found in the Materials and methods. (**C**) The flow cytometry results show that the knockdown of *GREB1*, *GHRHR* and *KLF8* inhibited *AR* reporter activity in LNCaP ARsig-hi cells. Top: The flow cytometry plot of one of the duplicate assays is shown. Bottom: The normalized median fluorescence intensity (MFI) of eGFP reporter in each assay is shown. *AR* shRNA was used as a positive control. *ACPP* shRNA is shown as a representative hairpin that had no effect on reporter activity. (**D**) The knockdown level of *AR*, *GREB1*, *GHRHR*, *KLF8* and *ACPP* from the cells represented in (**C**). The q-PCR data is presented as mean fold change ±SD relative to the shRenilla control. (**E**) The transcription of *GREB1*, *GHRHR* and *KLF8* is regulated by androgen in LNCaP. The data is presented as mean fold change ±SD relative to the DMSO control. NS = not significant, *p<0.05, **p<0.01, ***p<0.001, ****p<0.0001, one-way ANOVA compared to the DMSO control. (**F**) The correlation between RNA levels of *GREB1*, *GHRHR* and *KLF8* and AR score in 333 TCGA primary prostate tumors were analyzed using Pearson's correlation analysis (r). The RNA levels of the three genes were also compared between tumors with lowest (ARsig-lo, red points) and highest (ARsig-hi, blue points) AR score (5% of 333 cases: 17 cases each). NS = not significant, **p<0.01, ****p<0.0001, unpaired t-test. *One data point (*GHRHR*, x = −0.67, y = 1252.6072) is outside the y-axis limit. (**G**) The *GREB1* function is inhibited by CRISPR/Cas9 in four LNCaP ARsig-hi sublines. (Top) *AR* reporter activity was inhibited in all four *GREB1* CRISPR cell lines compared to control (SgNT). (Bottom) An

*Figure 2 continued on next page*

*Figure 2 continued*

example of the genomic alteration in the targeted sequence for each cell line is shown. (**H**) The CRISPR/Cas9-mediated inhibition of *GREB1* suppressed KLK3 expression without affecting the AR level.

DOI: https://doi.org/10.7554/eLife.41913.009

The following source data and figure supplement are available for figure 2:

**Source data 1.** Differentially expressed genes between ARsig-lo vs. ARsig-hi.

DOI: https://doi.org/10.7554/eLife.41913.011

**Source data 2.** Summary of Median eGFP Intensity of small-scale shRNA screen.

DOI: https://doi.org/10.7554/eLife.41913.012

**Source data 3.** AR scores and RNA levels of *GREB1*, *GHRHR* and *KLF8* of 333 TCGA cases.

DOI: https://doi.org/10.7554/eLife.41913.013

**Figure supplement 1.** Inhibition of *GREB1* suppresses *AR* transcriptional activity in CWR22Pc-EP cells.

DOI: https://doi.org/10.7554/eLife.41913.010

In breast cancer, *GREB1* functions as a coactivator through binding to ER and recruitment of the EP300/CBP complex to ER target genes (*Mohammed et al., 2013*). We find that *GREB1* functions similarly in prostate cells, as shown by co-immunoprecipitation documenting AR-GREB1 interaction (*Figure 3D*) and ChIP experiments showing recruitment of *GREB1* to *KLK3* and *FKBP5* enhancer regions (*Figure 3E*). Furthermore, ARsig-hi cells showed a *GREB1*-dependent increase in *EP300* binding (*Figure 3F,G*) and *GREB1* overexpression increased *EP300* recruitment to *AR* target genes in ARsig-lo cells (*Figure 3—figure supplement 2A*). Knockdown of *EP300* suppressed the effect of *GREB1* overexpression on DHT-induced *AR* target gene upregulation in ARsig-lo cells (*Figure 3—figure supplement 2B*, refer also to *Figure 3B*), suggesting that *EP300* is required for the function of *GREB1* as an *AR* co-factor.

In addition to this canonical coactivator function of promoting assembly of an active transcription complex, we found that *GREB1* also impacts *AR* DNA binding. For example, knockdown or CRISPR deletion of *GREB1* in ARsig-hi cells significantly reduced binding of *AR* to the *KLK3* enhancer and, conversely, *GREB1* overexpression promoted *AR* recruitment in ARsig-lo cells (*Figure 3H*, *Figure 3—figure supplement 2C*). *AR* ChIP-sequencing revealed that this effect is genome-wide, with a significant reduction in the mean height of *AR* peaks in *GREB1*-depleted cells (*Figure 3I–K*). Importantly, the location of *AR* peaks (enhancer, promoter) was identical in intact versus *GREB1* knockdown cells and there were no differences in consensus binding sites (*Figure 3—figure supplement 2D,E*). Therefore, *GREB1* enhances *AR* DNA efficiency but not alter DNA-binding site specificity. As seen previously in our analysis of ARsig-hi cells, total and nuclear *AR* levels were not changed by *GREB1* knockdown or overexpression (*Figure 3C*, *Figure 3—figure supplement 2F,G*).

Of note, earlier studies of *GREB1* in breast cancer did not report any effect on ER DNA binding (*Mohammed et al., 2013*), which we confirmed by *GREB1* knockdown in MCF7 breast cancer cells (*Figure 3—figure supplement 3A,B*). Thus, *GREB1* functions as a coactivator of both ER and AR but through somewhat different mechanisms. To address the possibility that other hormone receptor coactivators might also function differently in prostate cells, we asked if *NCOA1* and *NCOA2*, previously shown to recruit the EP300/CBP complex to *AR* (*Leo and Chen, 2000*), also influence *AR* DNA binding. To do so, we knocked down both genes in ARsig-hi cells based on prior work showing redundancy between *NCOA1* and *NCOA2* (*Leo and Chen, 2000*; *Wang et al., 2005*). *AR* reporter activity and target gene expression was inhibited in *NCOA1/2*-depleted cells, as expected, but *AR* occupancy of *AR* binding sites was unchanged (*Figure 3—figure supplement 3C–E*). Thus, in addition to a role in EP300/CBP recruitment, *GREB1* has unique effects on *AR* DNA binding that distinguish it from other coactivators.

## GREB1 is required for enzalutamide resistance of high AR output cells

Having demonstrated that *GREB1* is overexpressed in ARsig-hi cells and functions as an *AR* coactivator, we asked if *GREB1* is required for maintenance of the ARsig-hi state. First we evaluated the consequences of *GREB1* knockdown on transcription. Consistent with experiments in ARsig-lo cells showing that *GREB1* overexpression enhanced *AR* transcriptional activity (*Figure 3B,C*, *Figure 3—figure supplement 1B*), *GREB1* knockdown inhibited baseline and DHT-induced *AR* target gene expression in ARsig-hi cells (*Figure 4A–C*, *Figure 4—figure supplement 1A,B*). RNA-sequencing

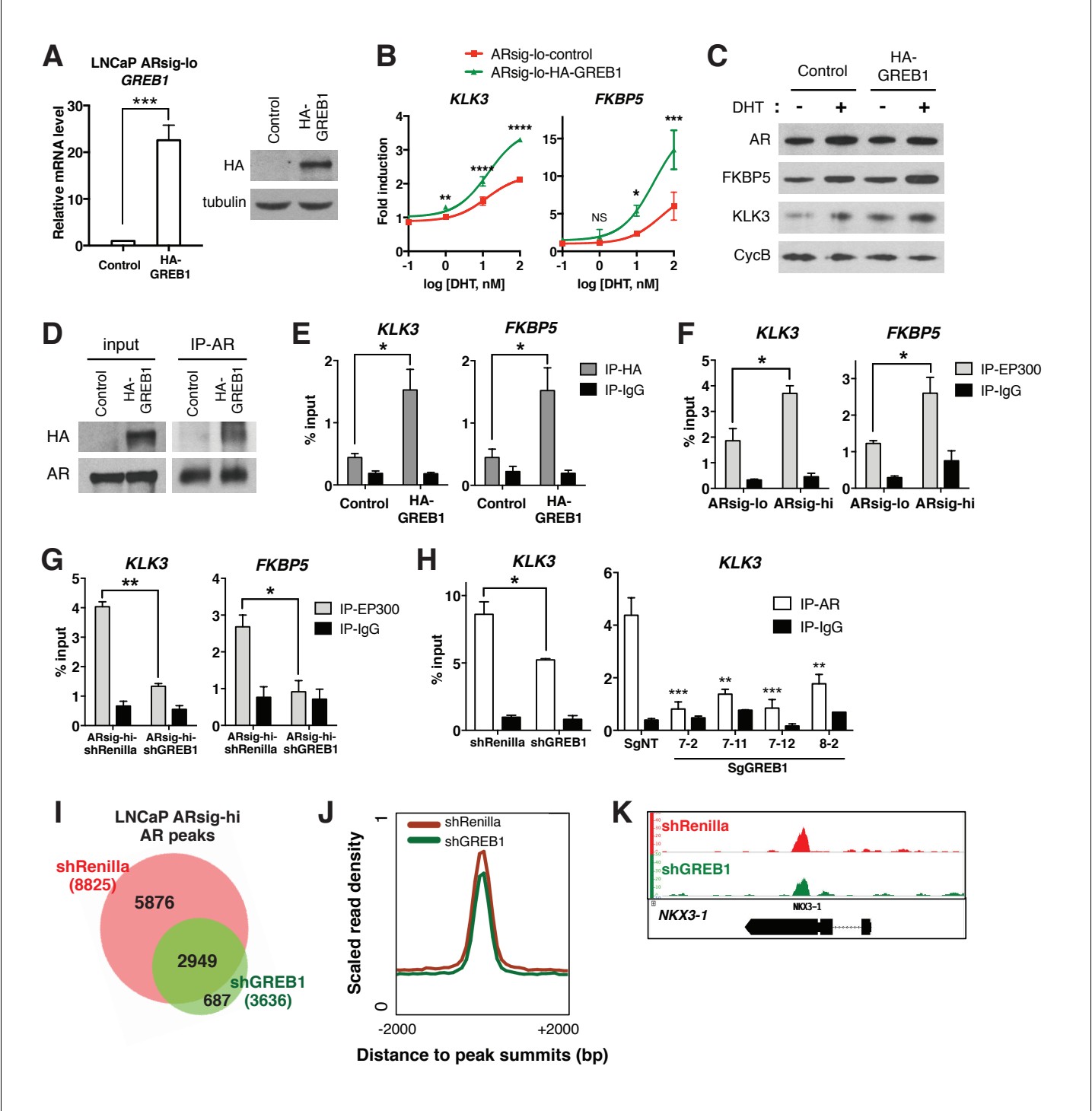

**Figure 3.** *GREB1* amplifies *AR* transcriptional activity by enhancing *AR* binding to chromatin. (**A**) *GREB1* overexpression in LNCaP ARsig-lo cells with stable integration of a *GREB1* lentiviral vector containing an amino-terminal HA-tag. (**B**) LNCaP ARsig-lo cells with *GREB1* overexpression show higher induction of *AR* target genes in response to DHT treatment. The q-PCR data is presented as mean fold change ±SD relative to the DMSO control. NS = not significant, *p<0.05, **p<0.01, ***p<0.001, ****p<0.0001, unpaired t-test compared to the control cells. (**C**) *GREB1* overexpression in LNCaP ARsig-lo cells increases protein levels of *AR* target genes without affecting *AR* level. (**D**) Co-immunoprecipitation using nuclear extracts shows an interaction between AR and GREB1 (HA) in LNCaP ARsig-lo cells. (**E**) ChIP against the HA-tag shows *GREB1* binding on the *KLK3* and *FKBP5* enhancer regions in LNCaP ARsig-lo cells. *p<0.05, unpaired t-test. (**F–G**) LNCaP ARsig-hi cells have increased *EP300* binding on the *KLK3* and *FKBP5* enhancer regions in a *GREB1* dependent manner. *p<0.05, **p<0.01, unpaired t-test. (**H**) *GREB1* knockdown or CRISPR decreases *AR* binding to *KLK3* enhancer

*Figure 3 continued on next page*

*Figure 3 continued*

in LNCaP ARsig-hi cells. *p<0.05, **p<0.01, ***p<0.001, unpaired t-test (shRenilla vs. shGREB1), one-way ANOVA (SgNT vs. SgGREB1). The ChIP q-PCR data (E–H) is presented as mean percentage input ±SD. (I) Overlap of *AR* ChIP-sequencing peaks shows that *AR* peaks are disrupted by *GREB1* knockdown in LNCaP ARsig-hi cells. (J) ChIP-sequencing summary plot shows that *AR* enrichment across the *AR*-binding sites is reduced by *GREB1* knockdown. (K) Example of AR genomic peaks at *NKX3-1*.

DOI: https://doi.org/10.7554/eLife.41913.014

The following figure supplements are available for figure 3:

**Figure supplement 1.** *GREB1* amplifies *AR* transcriptional activity in CWR22Pc-EP cells.

DOI: https://doi.org/10.7554/eLife.41913.015

**Figure supplement 2.** *GREB1* enhances *EP300* recruitment to *AR* and *AR* binding to chromatin.

DOI: https://doi.org/10.7554/eLife.41913.016

**Figure supplement 3.** *GREB1* has a unique function compared to ER or *NCOA1* and *NCOA2*.

DOI: https://doi.org/10.7554/eLife.41913.017

confirmed enrichment of androgen down-regulated gene sets in *GREB1*-depleted cells (*Figure 4D*) as well as downregulation of the 20 *AR* target genes used to calculate the *AR* activity score in TCGA tumors (*Figure 4—figure supplement 1C*). *GREB1* knockdown cells also showed enrichment of the same prostate basal gene set that was enriched in ARsig-lo cells (*Figure 4D*, refer also to *Figure 2A*). Additional analysis of RNA-seq data suggests that *GREB1* is a major molecular determinant of the ARsig-hi state: specifically, (i) *GREB1* knockdown impaired the induction of >70% of all DHT-induced genes (*Figure 4E*, *Figure 4—source datas 1* and *2*) and (ii) the top 100 gene sets enriched in *GREB1*-depleted ARsig-hi cells and ARsig-lo cells show significant overlap (*Figure 4F*, *Figure 4—source data 3*).

Earlier we showed that ARsig-hi cells rapidly acquire resistance to enzalutamide (refer to *Figure 1I*). To determine the role of *GREB1* in this drug resistant phenotype, we performed knockdown experiments using the LNCaP/AR xenograft. After confirming that *AR* activity was inhibited in ARsig-hi cells (*Figure 4—figure supplement 1D,E*), we injected LNCaP/AR ARsig-hi xenografts with *GREB1* shRNAs into castrated mice treated with enzalutamide and found a significant delay in the development of enzalutamide resistance after 10 weeks (*Figure 4G*). Clinical data from CRPC patients also supports for a role of *GREB1* in enzalutamide resistance. Although the samples are not matched pre- and post-treatment, we observed an overall increase in *GREB1* expression in those who progressed on enzalutamide treatment (*Figure 4H*). When we analyzed tumor purity content and stromal signature score as described previously (*Carter et al., 2012*; *Shah et al., 2017*; *Yoshihara et al., 2013*), no significant difference was observed between samples collected pre- vs. post-treatment (*Figure 4—figure supplement 1F*).

## Discussion

There is abundant evidence from tumor sequencing studies that genomic alterations in *AR* (amplification and/or mutation) are present in over 50% of CRPC patients (*Cancer Genome Atlas Research Network, 2015*; *Robinson et al., 2015*) and that *AR* amplification is associated with a less favorable clinical response to abiraterone or enzalutamide treatment (*Annala et al., 2018*). Therefore, high levels of *AR* transcriptional output can promote castration-resistant disease progression. Here we show that prostate cancers can amplify *AR* output through increased expression of the dual AR/ER coactivator *GREB1*, in the absence of genomic *AR* alterations. As with genomic *AR* amplification, increased *AR* output driven by high *GREB1* expression is also associated with enzalutamide resistance.

In addition to demonstrating the importance of transcriptional heterogeneity in drug resistance, we also show that *GREB1* amplifies *AR* activity by a novel two-part mechanism. Similar to canonical coactivators such as *NCOA1/2*, *GREB1* binds *AR* and promotes the assembly of an active transcription complex by recruitment of histone acetyl transferases such as EP300/CBP (*Lee and Lee Kraus, 2001*). However, *GREB1* has the additional property of improving the efficiency of *AR* binding to DNA, which further enhances *AR* transcriptional output. Although conceptually distinct from canonical coactivators, this dual mechanism of *AR* activation is may not be unique to *GREB1*. For example, *TRIM24* has been shown to function as an oncogenic *AR* cofactor and, similar to *GREB1*, knockdown of *TRIM24* impairs recruitment of *AR* to target genes (*Groner et al., 2016*). Curiously, the effect of

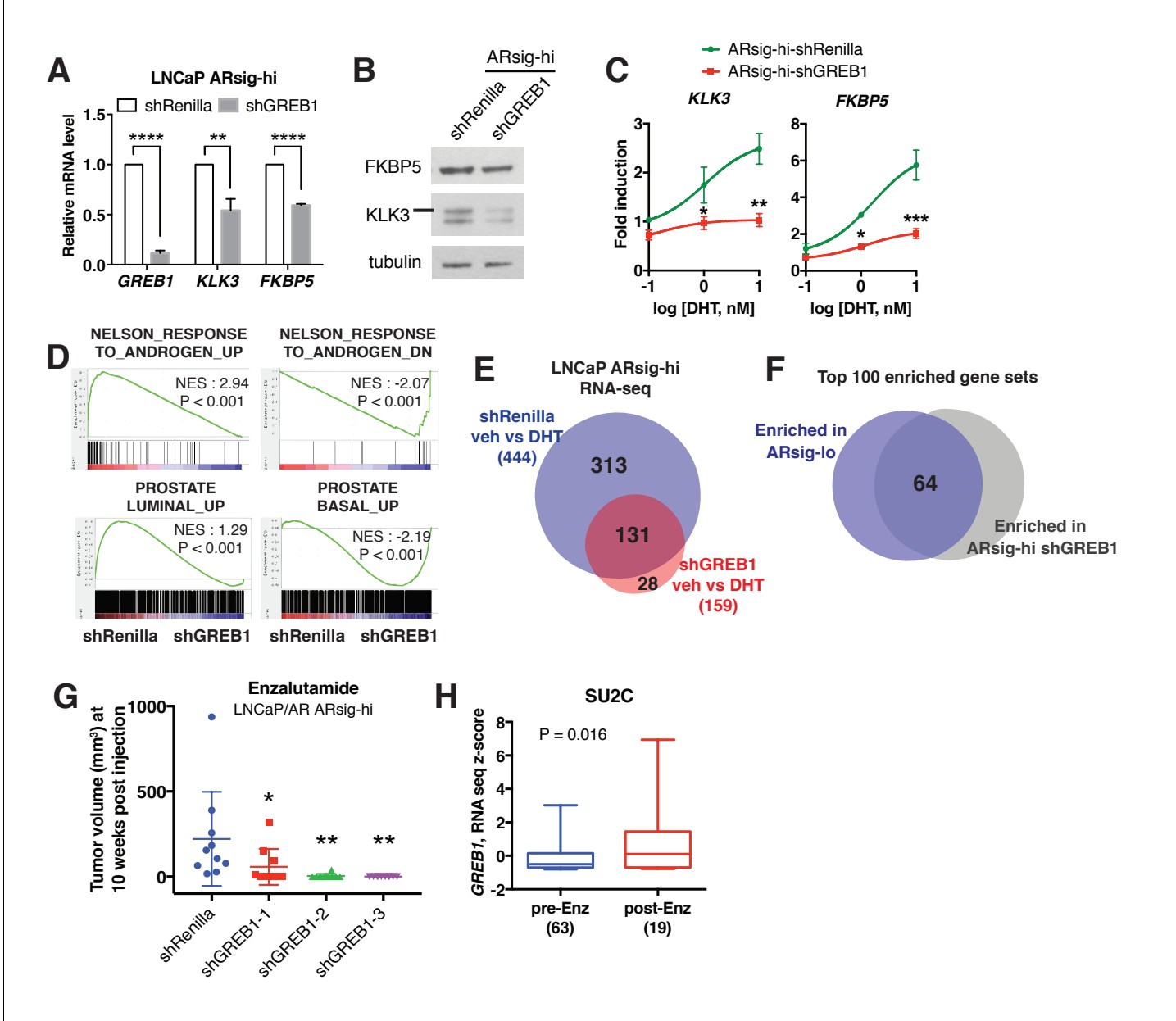

**Figure 4.** *GREB1* is the major molecular determinant of ARsig-hi cells. (A–B) Knockdown of *GREB1* inhibited *AR* target gene expression in LNCaP ARsig-hi cells. The q-PCR data (A) is presented as mean fold change ±SD relative to the shRenilla control. **p<0.01, ****p<0.0001, unpaired t-test. (C) Knockdown of *GREB1* suppressed the enhanced *AR* transcriptional activity in LNCaP ARsig-hi cells. The q-PCR data is presented as mean fold change ±SD relative to the DMSO control. *p<0.05, **p<0.01, ***p<0.001, unpaired t-test compared to the shRenilla control. (D) Gene set enrichment analysis (GSEA) shows that the gene sets up- and down-regulated by androgen are enriched in LNCaP ARsig-hi control and *GREB1* knockdown cells, respectively and genes upregulated in human prostate luminal and basal cells are enriched in ARsig-hi control and *GREB1* depleted cells, respectively. (E) Venn diagram showing that 70.5% of DHT-induced genes in control LNCaP ARsig-hi cells were inhibited by *GREB1* knockdown. (F) Venn diagram showing that 64% of the top 100 gene sets enriched in LNCaP ARsig-lo overlap with the top 100 gene sets enriched in *GREB1* depleted ARsig-hi cells. (G) Knockdown of *GREB1* inhibited development of enzlutamide-resistant LNCaP/AR xenografts derived from ARsig-hi cells. The sorted LNCaP/AR ARsig-hi cells were infected with control or three different shRNAs targeting *GREB1* and injected into physically castrated mice. Mice were treated with enzalutamide immediately after injection. Data is presented as mean ±SEM (N = 10). *p<0.05, **p<0.01, one-way ANOVA compared to the shRenilla control. (H) The SU2C cases that have received enzalutamide (Enz) have increased level of *GREB1* (unpaired t-test).

DOI: https://doi.org/10.7554/eLife.41913.018

The following source data and figure supplement are available for figure 4:

**Source data 1.** Upregulated genes in ARsig-hi shRenilla DHT vs. veh.

DOI: https://doi.org/10.7554/eLife.41913.020

*Figure 4 continued on next page*

*Figure 4 continued*

**Source data 2.** Upregulated genes in ARsig-hi shGREB1 DHT vs. veh.
DOI: https://doi.org/10.7554/eLife.41913.021
**Source data 3.** GSEA Results (ARsig-hi shRenilla DHT vs. shGREB1 DHT).
DOI: https://doi.org/10.7554/eLife.41913.022
**Figure supplement 1.** Knockdown of *GREB1* inhibits *AR* signaling.
DOI: https://doi.org/10.7554/eLife.41913.019

*GREB1* on *AR* DNA binding is not seen with ER, suggesting different conformational consequences of *GREB1* binding on AR and ER, respectively, then influence DNA binding.

One curious observation is the fact that prostate cancers can maintain transcriptional heterogeneity as a stable phenotype, despite the fact that *GREB1* expression drives a feed forward loop which, in principle, should result in an increased fraction of high *AR* output cells over time. One potential explanation for the ability of these populations to maintain stable proportions of high versus low *AR* output cells at steady state is the fact that androgen has growth inhibitory effects at higher concentrations (*Culig et al., 1999*). Because *GREB1* amplifies the magnitude of *AR* output in response to normal (growth stimulatory) androgen concentrations, the biologic consequence of high *GREB1* levels could be the same growth suppression seen with high androgen concentrations. This model predicts that high *AR* output cells would gain a fitness advantage under conditions of androgen deprivation or pharmacologic *AR* inhibition, as demonstrated by the enzalutamide resistance observed in xenograft models.

Further work is required to understand the clinical implications of our work, particularly whether *GREB1* levels in CRPC patients are predictive of response to next generation *AR* therapy. While we show that *GREB1* levels are elevated in the tumors of CRPC patients who have progressed on enzalutamide, it will be important to address this question prospectively, prior to next generation *AR* therapy. It is also important to note that the positive correlation of *GREB1* levels with high *AR* activity is largely based on the hormone-naïve TCGA cohort. It is also possible that the LNCaP cell line used for functional studies has an *AR* point mutation could potentially influence response to *GREB1* expression, but we obtained similar results in 22PC cells that lack this mutation (*Veldscholte et al., 1992*). In terms of therapeutic implications, *GREB1* knockdown experiments provide genetic evidence that *GREB1* is required for in vivo enzalutamide resistance in xenograft models. Although pharmacologic strategies to inhibit *GREB1* function are not currently available, a small molecule inhibitor that blocks protein-protein interactions between the AR N-terminal domain and CBP/EP300 is currently in clinical development (*Andersen et al., 2010*) (NCT02606123). This work provides precedent that similar strategies to disrupt GREB1/AR interaction may be possible.

## Materials and methods

**Key resources table**

| Reagent type (species) or resource | Designation | Source or reference | Identifiers | Additional information |
|---|---|---|---|---|
| Cell line (*H. sapiens*) | LNCaP | ATCC | CRL-1740, RRID:CVCL_1379 | |
| Cell line (*H. sapiens*) | LNCaP/AR | PMID: 14702632 | | |
| Cell line (*H. sapiens*) | CWR22Pc-EP | PMID: 28059768 | | |
| Antibody | AR | abcam | ab108341, RRID:AB_10865716 | WB (1:1000), IP (5 ug/IP) |
| Antibody | AR | Santa Cruz | sc-816, RRID:AB_1563391 | IF (1:500), ChIP (5 ug/IP) |
| Antibody | AR | Agilent | 441 | IHC |

*Continued on next page*

*Continued*

| Reagent type (species) or resource | Designation | Source or reference | Identifiers | Additional information |
|---|---|---|---|---|
| Antibody | KLK3 | Cell Signaling Technology | 5365 | WB (1:500) |
| Antibody | KLK3 | Biogenex | | IHC |
| Antibody | FKBP5 | Cell Signaling Technology | 8245, RRID:AB_10831198 | WB (1:500) |
| Antibody | TRPM8 | Epitomics | 3466–1, RRID:AB_10715643 | WB (1:1000) |
| Antibody | tubulin | Santa Cruz | sc-9104, RRID:AB_2241191 | WB (1:1000) |
| Antibody | Cyclophilin B | abcam | ab178397 | WB (1:100,000) |
| Antibody | BRD4 | Cell Signaling Technology | 13440, RRID:AB_2687578 | WB (1:1000) |
| Antibody | TOP2B | abcam | ab58442, RRID:AB_883147 | WB (1:1000) |
| Antibody | HA | Cell Signaling | 3724, RRID:AB_1549585 | WB (1:1000) |
| Antibody | HA | Abcam | ab9110, RRID:AB_307019 | ChIP (5 ug/IP) |
| Antibody | Alexa Fluor 647 | Invitrogen/ Thermo Fisher | A-31573, RRID:AB_2536183 | IF (1:1000) |
| Antibody | p300 | Santa Cruz | sc-585, RRID:AB_2231120 | ChIP (5 ug/IP) |
| Antibody | ER | Santa Cruz | sc-8002, RRID:AB_627558 | ChIP (5 ug/IP) |
| Antibody | normal rabbit IgG | Millipore Sigma | 12–370, RRID:AB_145841 | ChIP (5 ug/IP) |
| Antibody | Protein A/G agarose beads | Santa Cruz | sc-2003, RRID:AB_10201400 | |
| Recombinant DNA reagent | ARR3tk-eGFP/ SV40-mCherry | This paper | | Addgene plasmid #24304 |
| Recombinant DNA reagent | SCEP-shRenilla | This paper, PMID: 24332856 | | |
| Recombinant DNA reagent | SCEP-shAR.177 | This paper, PMID: 24332856 | | |
| Recombinant DNA reagent | SCEP-shGREB1-1 | This paper, PMID: 24332856 | | |
| Recombinant DNA reagent | SCEP-shGREB1-2 | This paper, PMID: 24332856 | | |
| Recombinant DNA reagent | SCEP-shGREB1-3 | This paper, PMID: 24332856 | | |
| Recombinant DNA reagent | SCEP-shKLF8.3467 | This paper, PMID: 24332856 | | |
| Recombinant DNA reagent | SCEP-shKLF8.2180 | This paper, PMID: 24332856 | | |
| Recombinant DNA reagent | SCEP-shKLF8.2684 | This paper, PMID: 24332856 | | |
| Recombinant DNA reagent | SCEP-shGHRHR.544 | This paper, PMID: 24332856 | | |
| Recombinant DNA reagent | SCEP-shGHRHR.1571 | This paper, PMID: 24332856 | | |
| Recombinant DNA reagent | SCEP-shGHRHR.1583 | This paper, PMID: 24332856 | | |

*Continued on next page*

*Continued*

| Reagent type (species) or resource | Designation | Source or reference | Identifiers | Additional information |
|---|---|---|---|---|
| Recombinant DNA reagent | SCEP-sh-p300-1 | This paper, PMID: 24332856 | | |
| Recombinant DNA reagent | SCEP-sh-p300-2 | This paper, PMID: 24332856 | | |
| Recombinant DNA reagent | SCEP-shSRC1-1 | This paper, PMID: 24332856 | | |
| Recombinant DNA reagent | SCEP-shSRC2-1 | This paper, PMID: 24332856 | | |
| Recombinant DNA reagent | SCEP-shSRC2-2 | This paper, PMID: 24332856 | | |
| Recombinant DNA reagent | lentiCRISPRv2-SgNT | PMID: 24336569 | | Addgene plasmid #52961 |
| Recombinant DNA reagent | lentiCRISPRv2-SgGREB1-7 | This paper | | Addgene plasmid #52961 |
| Recombinant DNA reagent | lentiCRISPRv2-SgGREB1-8 | This paper | | Addgene plasmid #52961 |
| Recombinant DNA reagent | pCMV6-GREB1 | PMID: 23403292 | | |
| Recombinant DNA reagent | pLVX-TRE3G-HA-GREB1 | This paper | | |
| Sequence-based reagent | q-PCR primers | This paper | | See *Supplementary file 1* |
| Sequence-based reagent | shRNAs | This paper | | |
| Sequence-based reagent | gRNAs | This paper | | |
| Commercial assay or kit | QIAshredder | Qiagen | 79656 | |
| Commercial assay or kit | RNeasy Mini Kit | Qiagen | 74106 | |
| Commercial assay or kit | High Capacity cDNA Reverse Transcription Kit | thermo fisher | 4368814 | |
| Commercial assay or kit | QuantiFast SYBR Green PCR Kit | Qiagen | 204057 | |
| Commercial assay or kit | BCA Protein Assay | ThermoFisher | 23225 | |
| Commercial assay or kit | Subcellular Protein Fractionation Kit | ThermoFisher | 78840 | |
| Commercial assay or kit | Peira TM900 system | Peira bvba | | |
| Commercial assay or kit | the KAPA Biosystems Hyper Library Prep Kit | Kapa Biosystems | KK8504 | |
| Chemical compound, drug | FBS | Omega Scientific | FB-11 | |
| Chemical compound, drug | Accumax | Innovative Cell Technologies | AM105 | |
| Chemical compound, drug | matrigel | Corning | 356237 | |
| Chemical compound, drug | Laemmli sample buffer | BioRad | 1610747 | |

*Continued on next page*

*Continued*

| Reagent type (species) or resource | Designation | Source or reference | Identifiers | Additional information |
|---|---|---|---|---|
| Chemical compound, drug | 4% formaldehyde | electron microscopy sciences | 15714 s | |
| Chemical compound, drug | normal goat serum | Vector Lab | S-1000, RRID:AB_2336615 | |
| Chemical compound, drug | normal horse serum | Vector Lab | S-2000, RRID:AB_2336617 | |
| Chemical compound, drug | 10% Triton X-100 solution | Teknova | T1105 | |
| Chemical compound, drug | DAPI mounting solution | Vector Lab | H-1200, RRID:AB_2336790 | |
| Chemical compound, drug | charcoal-stripped dextran-treated fetal bovine serum | Omega Scientific | FB-04 | |
| Chemical compound, drug | Puromycin | Invivogen | ant-pr | |
| Chemical compound, drug | RPMI | Media Preparation Core at Sloan Kettering Institute | | |
| Chemical compound, drug | DMEM | Media Preparation Core at Sloan Kettering Institute | | |
| Software, algorithm | Partek Genomics Suite software | Partek Inc | RRID:SCR_011860 | |
| Software, algorithm | FlowJo software | FlowJo software | RRID:SCR_008520 | version 9.9.6 |
| Software, algorithm | GSEA | Broad Institute | RRID:SCR_003199 | http://www.broad institute.org/ gsea/index.jsp |
| Software, algorithm | GraphPad Prism | GraphPad Prism | RRID:SCR_002798 | version 7 |
| Software, algorithm | STAR aligner | PMID: 23104886 | RRID:SCR_015899 | |
| Software, algorithm | Kalisto | PMID: 27043002 | | |
| Software, algorithm | RSeQC | PMID: 22743226 | RRID:SCR_005275 | http://broadinstitute .github.io/picard/ |
| Software, algorithm | DESeq2 package | http://www-huber. embl.de/users /anders/DESeq | RRID:SCR_015687 | |
| Software, algorithm | Picard | http://broadins titute.github.io/picard/index.html | RRID:SCR_006525 | |
| Software, algorithm | MACS2 | PMID: 22936215 | | |
| Software, algorithm | ChAsE | PMID: 27378294 | | |
| Software, algorithm | MEME-ChIP | PMID: 21486936 | | |
| Software, algorithm | HOMER | http://homer.ucsd. edu/homer/ | RRID:SCR_010881 | |

## Cell lines

LNCaP and MCF7 cell lines were obtained from American Type Culture Collection (ATCC, Manassas, VA) and maintained in RPMI (LNCaP) or DMEM (MCF7) +10% FBS (Omega Scientific, Tarzana, CA). LNCaP/AR cell line was generated and maintained as previously described (*Chen et al., 2004*).

CWR22Pc was a gift from Marja T. Nevalainen (Thomas Jefferson University, Philadelphia, PA) and CWR22Pc-EP was generated and maintained as previously described (*Mu et al., 2017*). Cell lines were authenticated by exome sequencing methods, and were negative for mycoplasma contamination testing.

## Flow cytometry analysis and FACS-sorting

Rapidly cycling eGFP *AR* reporter cells were collected using Accumax dissociation solution (Innovative Cell Technologies, San Diego, CA), and dead cells were counterstained with DAPI (Invitrogen, Grand Island, NY). For FACS-sorting of ARsig-lo and ARsig-hi cells, 5% of the entire population with lowest and highest eGFP expression was sorted out using BD FACSAria cell sorter. The 5% cutoff was used because it generates at least a 100-fold difference in median AR-GFP reporter signal between ARsig-lo and ARsig-hi cells and also allows us to have sufficient numbers of sorted cells to conduct various assays. For flow cytometric analysis of reporter activity, eGFP expression was measured using the BD-LDRII flow cytometer and analysis was done using FlowJo software.

## Plasmid construction and cell transduction

The lentiviral eGFP *AR* reporter (ARR$_3$tk-eGFP/SV40-mCherry) was generated by switching 7xTcf promoter of 7xTcf-eGFP/SV40-mCherry (Addgene, Cambridge, MA, 24304) with probasin promoter containing 3xARE (ARR$_3$tk) (*Snoek et al., 1998*). For shRNA knockdown experiments, SCEP vector was generated by substituting GFP cassette of SGEP (pRRL-GFP-miRE- PGK-PuroR, gift from Johannes Zuber) (*Fellmann et al., 2013*) with mCherry cassette. The following guide sequences were used for knockdown:

    shAR.177: TAGTGCAATCATTTCTGCTGGC
    shGREB1-1: TTGTCAGGAACAGACACTGGTT
    shGREB1-2: TTTCAGATTTATATGATTGGAG
    shGREB1-3: TTGACAAGATACCTAAAGCCGA
    shKLF8.3467: TTGAGTTCTAAAGTTTTCCTGA
    shKLF8.2180: TATTTGTCCAAATTTAACCTAA
    shKLF8.2684: TTATAAACAATCTGATTGGGC
    shGHRHR.544: TAAAAGTGGTGAACAGCTGGGT
    shGHRHR.1571: TTTATTGGCTCCTCTGAGCCTT
    shGHRHR.1583: TTCATTTACAGGTTTATTGGCT
    shEP300-1: TCCAGAAAGAACTAGAAGAAAA
    shEP300-2: TTAATCTATCTTCAGTAGCTTG
    shNCOA1-1: TTCTTCTTGGAACTTGTCGTTT
    shNCOA2-1: TTGCTGAACTTGCTGTTGCTGA
    shNCOA2-2: TTAACTTTGCTCTTCTCCTTGC

shRenilla was previously described as Ren.713 targeting Renilla luciferase (*Fellmann et al., 2013*). Pools of 3 shRNAs were used to knockdown *GREB1*, *KLF8* and *GHRHR* in a small-scale shRNA screen, and shGREB1-1 was used for further studies. For CRISPR/Cas9 experiments, lentiCRISPRv2 vector gifted by F. Zhang (Addgene, 52961) was used with the following guide sequences designed using http://crispr.mit.edu/ website:

    SgGREB1-7: AGGCATGTCCTGCGTGCCGC
    SgGREB1-8: TCACGGGCATACGAGCAGTA sgNT was previously described (*Wang et al., 2014*).

pCMV6-GREB1 plasmid was a gift from J. Carroll (Cancer Research UK Cambridge Institute, Cambridge, UK). The lentiviral *GREB1* cDNA plasmid was constructed by cloning *GREB1* cDNA from pCMV6-GREB1 into Tet-inducible pLV-based lentiviral expression vector with HA-tag.

Lentiviral transduction of cells was performed as described previously (*Mu et al., 2017*). To make *AR* reporter cell line, cells were infected with ARR$_3$tk-eGFP/SV40-mCherry at low multiplicity of infection (MOI) to enable each cell has one copy of reporter construct, and the transduced cells were sorted by mCherry flow cytometry. To inactivate *GREB1* gene, we single-cell cloned the cells infected with lentiCRISPRv2 vector containing SgGREB1-7 or SgGREB1-8, and isolated a clone that had genomic alteration at target sequence. Three clones were generated by using SgGREB1-7 (SgGREB1-7-2, 7–11 and 7–12) and one clone was generated by using SgGREB1-8 (SgGREB1-8-2).

## shRNA screen

FACS-based small-scale shRNA screen with 33 selected genes was performed as follows: FACS-sorted ARsig-hi cells were plated in 12 well plate ($1.5 \times 10^5$ cells per well, Corning, 353043) and each well was infected with pool of 3 SEPC shRNAs against each gene on the following day. Cells with stable integration of hairpins were selected with 2 μg/ml puromycin. 9 days after infection, half of the cells in each well was used to analyze eGFP *AR* reporter activity using flow cytometry, and the other half was subjected to qRT-PCR to determine knockdown level of the gene. We performed the screen in duplicate and each replicate included wells infected with shRenilla or shAR as controls. The median fluorescence intensity (MFI) of eGFP was measured using FlowJo software. The shRNAs decreased eGFP MFI more than 1.5 fold compared to shRenilla (normalized value lower than 0.667) in both duplicate were considered as hits. The list of 33 genes used in the screen and the summary of median eGFP intensity can be found at *Figure 2—source data 2*.

## Xenograft assay

To compare time to acquire enzalutamide resistance in vivo, FACS-sorted bulk, ARsig-lo and ARsig-hi populations derived from LNCaP/AR were cultured for 6 days after sorting to obtain enough number of cells for xenograft assay. $2 \times 10^6$ cells were injected subcutaneously into the flank of physically castrated CB17 SCID mice in a 50:50 mix of matrigel (BD Biosciences, San Jose, CA) and regular culture medium (five mice, 10 tumors per group), and enzalutamide treatment was initiated on the day of injection. To test the effect of GREB1 knockdown on development of enzalutamide resistance, FACS-sorted ARsig-hi population derived from LNCaP/AR was infected with control or three different shGREB1 constructs 2 days after sorting. Cells with stable integration of hairpin were selected with 2 μg/ml puromycin. 5 days after infection, $2 \times 10^6$ cells were injected subcutaneously into the flank of castrated CB17 SCID mice (five mice, 10 tumors per group), and enzalutamide treatment was initiated on the day of injection. The same cell populations used for injection were also used to test eGFP *AR* reporter activity using flow cytometry, and qRT-PCR to test knockdown level of *GREB1*. Measurements were obtained weekly using Peira TM900 system (Peira bvba, Belgium). All animal experiments were performed in compliance with the approved institutional animal care and use committee (IACUC) protocols (#06-07-012) of the Research Animal Resource Center of Memorial Sloan Kettering Cancer Center.

## Immunoblot, immunoprecipitation and immunostaining

Protein was extracted from cells using Triton lysis buffer and quantified by BCA Protein Assay (ThermoFisher Scientific, Waltham, MA, 23225). Nuclear/cytoplasmic fractionation was achieved with Subcellular Protein Fractionation Kit (ThermoFisher Scientific, 78840). Protein lysates were subjected to SDS-PAGE and immunoblotted with the following antibodies against: AR (Abcam, Cambridge, United Kingdom, ab108341), KLK3 (Cell Signaling Technology, Danvers, MA, 5365), FKBP5 (Cell Signaling, 8245) TRPM8 (Epitomics, Burlingame, CA, 3466–1), tubulin (Santa Cruz Biotechnology, Dallas, TX, sc-9104), Cyclophilin B (Abcam, ab178397), BRD4 (Cell Signaling, 13440), TOP2B (Abcam, ab58442), HA (Cell Signaling, 3724).

For AR immunoprecipitation, at least 1.5 mg of total protein was incubated with AR antibody (Abcam, ab108341) overnight at 4°C followed by the addition of Protein A/G agarose beads (Santa Cruz, sc-2003) for 2 hr. Immune complexes were extensively washed with Triton buffer and solubilized using Laemmli sample buffer (BioRad, Hercules, CA).

For immunofluorescence staining, cells were fixed with 4% formaldehyde, permeabilized with 0.2% Triton-X, blocked with 5% normal goat and 5% normal horse serum, stained with anti-AR (Santa Cruz, sc-816) primary and Alexa Fluor 647 (Invitrogen) secondary antibodies, and mounted with DAPI mounting solution (Vector Lab, Burlingame, CA). For Immunohistochemistry, tumor sections were stained with anti-AR (Agilent, Santa Clara, CA, 441) and KLK3 (Biogenex, Fremont, CA) antibodies using Leica Bond RX (Leica Biosystems, Wetzlar, Germany).

## Transcription analysis

Total RNA was isolated using the QiaShredder kit (Qiagen, Germantown, MD) for cell lysis and the RNeasy kit (Qiagen) for RNA purification. For quantitative PCR with reverse transcription (RT–qPCR), we used the High-Capacity cDNA Reverse Transcription Kit (Applied Biosystems, Grand Island, NY)

to synthesize cDNA according to the manufacturer's protocol. Real-time PCR was performed using gene-specific primers and 2X SYBR green quantfast PCR Mix (Qiagen, 1044154). Data were analyzed by the DDCT method using GAPDH as a control gene and normalized to control samples, which were arbitrarily set to 1. To test DHT-induced *AR* target gene upregulation, cells were hormone-deprived in 10% charcoal-stripped dextran-treated fetal bovine serum (Omega Scientific) media for 2 days and then treated with indicated concentration of DHT for 24 hr. Triplicate measurements were made on at least three biological replicates. The primer sequences used for q-PCR are listed at *Supplementary file 1*.

For RNA-seq, library preparation, sequencing and expression analysis were performed by the New York Genome Center. Libraries were prepared using TruSeq Stranded mRNA Library Preparation Kit in accordance with the manufacturer's instructions and sequenced on an Illumina HiSeq2500 sequencer (rapid run v2 chemistry) with 50 base pair (bp) reads. Partek Genomics Suite software (Partek Inc, St. Louis, MO) was used to analyze differentially expressed genes between ARsig-lo vs. ARsig-hi (Fold change $\geq$1.5, p<0.05). To analyze RNA-seq data from ARsig-hi cells with shRenilla vs. shGREB1, reads were aligned to the NCBI GRCh37 human reference using STAR aligner (*Dobin et al., 2013*). Quantification of genes annotated in Gencode vM2 were performed using featureCounts and quantification of transcripts using Kalisto (*Bray et al., 2016*). QC were collected with Picard and RSeQC (*Wang et al., 2012*) (http://broadinstitute.github.io/picard/). Normalization of feature counts was done using the DESeq2 package (http://www-huber.embl.de/users/anders/DESeq/). Differentially expressed genes were defined as a 1.5 fold difference, p<0.05 of DESeq-normalized expression. For GSEA, statistical analysis was performed with publicly available software from the Broad Institute (http://www.broadinstitute.org/gsea/index.jsp). The basal and luminal gene signatures used for GSEA (*Supplementary file 2*) were generated by conducting RNA-sequencing with normal human basal vs. luminal prostate cells isolated as previously described (*Karthaus et al., 2014*). Full description of this study will be reported separately.

## ChIP

ChIP experiments were performed as previously described (*Arora et al., 2013*), using SDS-based buffers. Antibodies were used at a concentration of 5 ug per 1 mL of IP buffer, which encompassed approximately 8 million cells per IP. Antibodies used were: AR (Santa Cruz, sc-816), EP300 (Santa Cruz, sc-585), HA (Abcam, ab9110), ER (Santa Cruz, sc-8002). The primer sequences used for ChIP-qPCR are listed at *Supplementary file 1*.

For ChIP–seq, library preparation and RNA-seq were performed by the NYU Genome Technology Center. Libraries were made using the KAPA Biosystems Hyper Library Prep Kit (Kapa Biosystems, Woburn, MA, KK8504), using 10 ng of DNA as input and 10 PCR cycles for library amplification. The libraries were sequenced on a HiSeq 2500, as rapid run v2 chemistry, paired-end mode of 51 bp read length.

The ChIP-seq reads were aligned to the human genome (hg19, build 37) using the program BWA (VN: 0.7.12; default parameters) within the PEMapper. Duplicated reads were marked by the software Picard (VN: 1.124; http://broadinstitute.github.io/picard/index.html) and removed. The software MACS2 (*Feng et al., 2012*) (-q 0.1) was used for peak identification with data from ChIP input DNAs as controls. Peaks of sizes > 100 bp and with at least one base pair covered by >18 reads were selected as the final high confident peaks. Peaks from shGREB1/control conditions were all merged to obtain non-overlapping genomic regions, which were then used to determine conditional specific *AR* binding. Overlapped peaks were defined as those sharing at least one base pair. To generate graphs depicting *AR* ChIP–seq read density in ±2 kilobase regions of the *AR* peak summits, the same number of ChIP–seq reads from different conditions were loaded into the software ChAsE (*Younesy et al., 2016*), and the resulting read density matrices were sorted by the read densities in the shRenilla control, before coloring. The read density was also used to select peaks with significant signal difference between shGREB1 and controls. The criteria for assigning peaks to genes have been described previously (*Rockowitz and Zheng, 2015*). The MEME-ChIP software (*Machanick and Bailey, 2011*) was applied to 300 bp sequences around the peak summits for motif discovery, and the comparison of sequence motifs was also analyzed with HOMER (http://homer.ucsd.edu/homer/).

## Analysis of human prostate cancer datasets

All analysis of human prostate cancer data was conducted using previously published datasets of The Cancer Genome Atlas (TCGA) (*Cancer Genome Atlas Research Network, 2015*) and PCF/SU2C (*Robinson et al., 2015*), which can be explored in the cBioPortal for Cancer Genomics (http://www.cbioportal.org). Tumor purity content was estimated computationally using the ABSOLUTE method (*Carter et al., 2012*), based on mutant allele variant fractions and zygosity shifts. Stromal signature score was applied to the normalized RNA-seq expression dataset (*Yoshihara et al., 2013*).

## Statistics

For comparison of pooled data between two different groups, unpaired t tests were used to determine significance. For comparison of data among three groups, one-way ANOVA was used to determine significance. In vitro assays represent three independent experiments from biological replicates, unless otherwise indicated. In all figures, $*p<0.05$, $**p<0.01$, $***p<0.001$ and $****p<0.0001$. For GSEA, statistical analysis was performed with publicly available software from the Broad Institute (http://www.broadinstitute.org/gsea/index.jsp). The sample size estimate was based on our experience with previous experiments (*Balbas et al., 2013*; *Bose et al., 2017*; *Chen et al., 2013*). No formal randomization process was used to assign mice to a given xenograft assay, and experimenters were not blinded.

## Acknowledgements

We thank the flow cytometry core facility at MSKCC for technical support, NYU Genome Technology Center for conducting ChIP-sequencing, New York Genome Center for conducting the RNA-sequencing, MSKCC Pathology Core for assistance with IHC staining of patient samples, Wouter Karthaus for help with cloning and providing the basal and luminal gene signature, Wassim Abida for help with analyzing patient data, Kayla Lawrence and Tejasveeta Nadkarni for help with cloning and Jason Carroll (Cancer Research UK Cambridge Institute) for generously providing pCMV6-GREB1 plasmid, and the members of the Sawyers laboratory for helpful discussions.

## Additional information

### Competing interests

Charles L Sawyers: Senior Editor eLife; Board of Directors of Novartis; co-founder of ORIC Pharm; co-inventor of enzalutamide and apalutamide; Science advisor to Agios, Beigene, Blueprint, Column Group, Foghorn, Housey Pharma, Nextech, KSQ, Petra and PMV; co-founder of Seragon, purchased by Genentech/Roche in 2014. John Wongvipat: co-inventor of enzalutamide. The other authors declare that no competing interests exist.

### Funding

| Funder | Grant reference number | Author |
|---|---|---|
| U.S. Department of Defense | W81XWH-15-1-0540 | Eugine Lee |
| Iris and Junming Le Foundation | | Eugine Lee |
| Howard Hughes Medical Institute | | Charles L Sawyers |
| National Institutes of Health | CA155169 | Charles L Sawyers |
| Starr Foundation | I10-0062 | Charles L Sawyers |
| National Institutes of Health | CA193837 | Charles L Sawyers |
| National Institutes of Health | CA224079 | Charles L Sawyers |
| National Institutes of Health | CA092629 | Charles L Sawyers |
| National Institutes of Health | CA160001 | Charles L Sawyers |

| National Institutes of Health | CA008748 | Charles L Sawyers |

The funders had no role in study design, data collection and interpretation, or the decision to submit the work for publication.

## Author contributions

Eugine Lee, Conceptualization, Data curation, Formal analysis, Funding acquisition, Validation, Investigation, Visualization, Methodology, Writing—original draft, Project administration, Writing—review and editing; John Wongvipat, Formal analysis, Investigation, Methodology; Danielle Choi, Formal analysis, Investigation; Ping Wang, Formal analysis, Visualization; Young Sun Lee, Investigation; Deyou Zheng, Formal analysis, Visualization, Writing—review and editing; Philip A Watson, Resources, Investigation, Writing—review and editing; Anuradha Gopalan, Resources; Charles L Sawyers, Conceptualization, Supervision, Funding acquisition, Writing—original draft, Project administration, Writing—review and editing

## Author ORCIDs

Deyou Zheng (iD) http://orcid.org/0000-0003-4354-5337
Charles L Sawyers (iD) http://orcid.org/0000-0003-4955-6475

## Ethics

Animal experimentation: All animal experiments were performed in compliance with the approved institutional animal care and use committee (IACUC) protocols (#06-07-012) of the Research Animal Resource Center of Memorial Sloan Kettering Cancer Center.

## Decision letter and Author response

Decision letter https://doi.org/10.7554/eLife.41913.035
Author response https://doi.org/10.7554/eLife.41913.036

# Additional files

## Supplementary files

• Supplementary file 1. Primer list.
DOI: https://doi.org/10.7554/eLife.41913.023
• Supplementary file 2. The basal and luminal gene signatures used for GSEA.
DOI: https://doi.org/10.7554/eLife.41913.024
• Transparent reporting form
DOI: https://doi.org/10.7554/eLife.41913.025

## Data availability

RNA-seq data has been deposited in GEO under accession code GSE120720. ChIP-seq data has been deposited in GEO under accession code GSE120680

The following datasets were generated:

| Author(s) | Year | Dataset title | Dataset URL | Database and Identifier |
|---|---|---|---|---|
| Lee E, Wongvipat J, Choi D, Wang P, Lee YS, Zheng D, Watson PA, Gopalan A, Sawyers CL | 2019 | GREB1 amplifies androgen receptor output in prostate cancer and contributes to antiandrogen resistance | https://www.ncbi.nlm.nih.gov/geo/query/acc.cgi?acc=GSE120720 | NCBI Gene Expression Omnibus, GSE120720 |
| Lee E, Wongvipat J, Choi D, Wang P, Lee YS, Zheng D, Watson PA, Gopalan A, Sawyers CL | 2019 | GREB1 amplifies androgen receptor output in prostate cancer and contributes to antiandrogen resistance | https://www.ncbi.nlm.nih.gov/geo/query/acc.cgi?acc=GSE120680 | NCBI Gene Expression Omnibus, GSE120680 |

The following previously published datasets were used:

| Author(s) | Year | Dataset title | Dataset URL | Database and Identifier |
|---|---|---|---|---|
| Robinson D, Van Allen EM, Wu YM, Schultz N, Lonigro RJ, Mosquera JM, Montgomery B, Taplin ME, Pritchard CC, Attard G, Beltran H, Abida W, Bradley RK, Vinson J, Cao X, Vats P, Kunju LP, Hussain M, Feng FY, Tomlins SA, Cooney KA, Smith DC, Brennan C, Siddiqui J, Mehra R, Chen Y, Rathkopf DE, Morris MJ, Solomon SB | 2015 | Integrative clinical genomics of advanced prostate cancer | https://www.ncbi.nlm.nih.gov/gap/?term=phs000915.v1.p1 | NCBI dbGap, phs000915.v1.p1 |
| Cancer Genome Atlas Research Network | 2015 | The Molecular Taxonomy of Primary Prostate Cancer | http://www.cbioportal.org/study.do?cancer_study_id=prad_tcga_pub | cBioPortal for Cancer Genomics, prad_tcga_pub |

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
