## [Decision Letter]

Thank you for submitting your article "*GREB1* amplifies androgen receptor output in prostate cancer and contributes to antiandrogen resistance" for consideration by *eLife*. Your article has been reviewed by three peer reviewers, and the evaluation has been overseen by a Reviewing Editor and Sean Morrison as the Senior Editor. The following individuals involved in review of your submission have agreed to reveal their identity: Gert Attard (Reviewer #2); Myles Brown (Reviewer #3).

The reviewers have discussed the reviews with one another and the Reviewing Editor has drafted this decision to help you prepare a revised submission.

Summary:

The finding that *GREB1* amplifies androgen output in prostate cancer cell lines and does so by 2 mechanisms including enhancing AR binding and co-activator activity has implications for resistance of PC to enzalutamide and other AR targeting agents.

Essential revisions:

1) Statistical tests/results should be added to the figure panels and should be more clearly specified throughout the text and in the figures.

2) The authors use both LNCAP and LNCaP with stably over-expressing AR (for xenograft work and based on previous studies). The use of these models needs to be clarified in the figures (especially Figure 1) as it is currently hard to follow.

3) Figure 2F. Please indicate what AR score was used and what is the y axis? Ideally the distribution for *GREB1* expression across the full data set should be shown and the ARsig-hi and ARsig-lo marked on this?

4) Regarding Figure 4H, in the Discussion the authors conclude "we show that elevated levels of *GREB1* in CRPC tumors correlate with a poor clinical response to enzalutamide, analogous to the prognostic impact of AR gene amplification". This is inaccurate. The authors show increased expression in tumors collected at progression vs. baseline. Have these been controlled for tumor fraction and other potential biases?

5) There are a few grammatical errors in the figure legends, such as in that of Figure 4H. Please correct.

---

## [Author Response]

Essential revisions:1) Statistical tests/results should be added to the figure panels and should be more clearly specified throughout the text and in the figures.

Thank you for noting this. We have added statistical results in revised Figure 1D, 1H, 1I, Figure 1—figure supplement 1B, 1E, Figure 1—figure supplement 3B, Figure 2E, Figure 2—figure supplement 1A, Figure 3A, 3B, 3E, 3F, 3G, 3H, Figure 3—figure supplement 1A, 1B, Figure 3—figure supplement 2A, 2C, Figure 3—figure supplement 3A, 3B, 3D, 3E, Figure 4A, 4C, Figure 4—figure supplement 1A and 1E. We have also edited the figure legends to specify the statistical analysis used in each figure.

2) The authors use both LNCAP and LNCaP with stably over-expressing AR (for xenograft work and based on previous studies). The use of these models needs to be clarified in the figures (especially Figure 1) as it is currently hard to follow.

We appreciate the feedback and have specified the cell lines throughout in the figures and figure legends.

3) Figure 2F. Please indicate what AR score was used and what is the y axis? Ideally the distribution for GREB1 expression across the full data set should be shown and the ARsig-hi and ARsig-lo marked on this?

We used the same AR score as in the TCGA study (Cancer Genome Atlas Research, 2015; Hieronymus et al., 2006). The original Figure 2F did not have y-axis. As suggested by the reviewers, we’ve combined the original Figure 2F and 2G and now show the distribution of *GREB1, GHRHR* and *KLF8* expression across the 333 TCGA tumors. We have marked ARsig-lo and ARsig-hi cases to show that the expression of *GREB1*, but not *GHRHR* or *KLF8*, is increased in ARsig-hi cases (new Figure 2F). We have also analyzed the Pearson’s correlation (*r*) between RNA level of each gene and AR score across the TCGA tumors and observed that only *GREB1* expression shows significantly positive correlation with AR score (*r*=0.55, P<0.0001).

4) Regarding Figure 4H, in the Discussion the authors conclude "we show that elevated levels of GREB1 in CRPC tumors correlate with a poor clinical response to enzalutamide, analogous to the prognostic impact of AR gene amplification". This is inaccurate. The authors show increased expression in tumors collected at progression vs baseline. Have these been controlled for tumor fraction and other potential biases?

We thank the reviewers for pointing this out and have edited the Discussion to clarify this. We did not intend to claim that *GREB1* is a prognostic biomarker (although future work may address this possibility).

Regarding tumor purity, we have analyzed tumor content and stromal signature scores of patient samples as described previously (Carter et al., 2012; Shah et al., 2017; Yoshihara et al., 2013). We observed no significant change in both tumor purity content and stromal signature score between pre- vs. post-enzalutamide setting, suggesting that increased *GREB1* expression is not caused by different tumor content between samples. These analyses are presented in new Figure 4—figure supplement 1F.

5) There are a few grammatical errors in the figure legends, such as in that of Figure 4H. Please correct.

We appreciate the feedback and have corrected grammatical errors in figure legends.